# Emerging Technologies and Open-Source Platforms for Remote Physical Exercise: Innovations and Opportunities for Healthy Population—A Narrative Review

**DOI:** 10.3390/healthcare12151466

**Published:** 2024-07-23

**Authors:** Alberto Fucarino, Antonio Fabbrizio, Nuno D. Garrido, Enzo Iuliano, Victor Machado Reis, Martina Sausa, José Vilaça-Alves, Giovanna Zimatore, Carlo Baldari, Filippo Macaluso, Andrea De Giorgio, Manuela Cantoia

**Affiliations:** 1Department of Theoretical and Applied Sciences, eCampus University, 22060 Novedrate, Italy; alberto.fucarino@uniecampus.it (A.F.); antonio.fabbrizio@uniecampus.it (A.F.); enzo.iuliano@uniecampus.it (E.I.); martina.sausa@gmail.com (M.S.); giovanna.zimatore@uniecampus.it (G.Z.); carlo.baldari@uniecampus.it (C.B.); andrea.degiorgio@uniecampus.it (A.D.G.); manuela.cantoia@uniecampus.it (M.C.); 2Research Center in Sports Sciences, Health Sciences and Human Development, CIDESD, 5000-801 Vila Real, Portugal; nggarrido@utad.pt (N.D.G.); vmreis@utad.pt (V.M.R.); josevilaca@utad.pt (J.V.-A.); 3Sciences Departament, University of Tra’s-os-Montes e Alto Douro, 5000-801 Vila Real, Portugal

**Keywords:** healthy lifestyle, physical activity, tele-exercise, wellness, wellbeing

## Abstract

The emergence of tele-exercise as a response to the impact of technology on physical activity has opened up new possibilities for promoting physical health. By integrating innovative technologies and open-source platforms, tele-exercise encourages people to stay active. In our latest analysis, we delved into the scientific literature surrounding the use of tele-exercise technologies in training healthy individuals. After conducting an extensive search on the PubMed database using the keywords “tele-exercise” and “physical activity” (from 2020 to 2023), we identified 44 clinical trials that were applicable to tele-exercise, but less than 10% of them were aimed at healthy individuals, precisely 9.09% (four out of forty-four studies analyzed). Our review highlights the potential of tele-exercise to help maintain physical fitness and psychological well-being, especially when traditional fitness facilities are not an option. We also underscore the importance of interoperability, standardization, and the incorporation of biomechanics, exercise physiology, and neuroscience into the development of tele-exercise platforms. Nevertheless, despite these promising benefits, research has shown that there is still a significant gap in the knowledge concerning the definition and evaluation of training parameters for healthy individuals. As a result, we call for further research to establish evidence-based practices for tele-exercise in the healthy population.

## 1. Introduction

In today’s world of rapid technological evolution and democratized information access, tele-exercise is a rapidly developing field [1]. Tele-exercise, also known as “telefitness,” is a cutting-edge combination of computer engineering and exercise science. This union symbolizes the inevitable fusion of technological innovation and the human imperative to preserve an active and healthy lifestyle. Recent technological evolutions, particularly advanced digitization capabilities and the expansion of the Internet of Things (IoT) [2,3], have fostered the formation of an ecosystem in which hardware (such as wearables, motion sensors, and smart workout equipment) and software (including mobile applications, cloud solutions, and advanced data analytics algorithms) [4] work together symbiotically to revolutionize the workout experience [5].

Open-source platforms have become significant players in this field. In contrast to proprietary software solutions, open source allows for the free distribution of source code, providing an unrestricted environment for collaboration, innovation, and adaptability. Open-source platforms are transforming the development and deployment of fitness solutions due to their collaborative, flexible, and open nature [6]. These platforms enable quick identification and bug fixing through rapid iteration, along with customized solutions for specific members of the population [7]. Further support for the benefit of tele-exercise activities is provided by new technologies developed in the area of wearable devices. Wearable devices connected to advanced networks facilitate the real-time collection of biomechanical and physiological data. Analyzing these data through open-source platforms provides immediate feedback to the user, ensuring personalized training [8]. Integration with other systems, such as health monitoring applications, virtual/augmented reality platforms, and dietary applications, has become more seamless [9]. Virtual reality (VR) is emerging as a powerful immersion tool. With advanced graphics processing techniques and virtual training environments, it can respond dynamically to the user’s performance. In addition, through integration with motion capture systems, VR can assist rehabilitation [10], while gamification dynamics encourage the user to maintain consistent engagement [11]. Augmented reality (AR) technology, through the use of advanced optical tracking and inertial measurement unit sensors, enables precise fusion between real and virtual elements, providing immediate feedback on user posture and form during exercises. Integration with adversarial generative networks and encrypted peer-to-peer networks allows for dynamic scenarios and secure performance sharing [12]. Artificial intelligence (AI) is also playing an increasingly central role [13]: using deep neural networks [14] and support vector machines, tailored training programs can be generated based on an advanced classification of the user’s physical performance. In addition, convolutional neural networks, suitable for time-sequence analysis, offer real-time recommendations, while algorithms such as k-means clustering can identify specific training phases [15]. Tele-exercise platforms often leverage AR and VR to provide immersive training [16].

The engineering capabilities of new generation platforms are [17]:Modularity and Interoperability: integrating various devices and services is facilitated by open-source frameworks’ modular structure;Scalability: the barrier-free collaboration inherent in open source can lead to more efficient algorithms, enabling systems to handle increasing amounts of data and users;Advanced Customization: access to source code allows for deep customization, enabling specific changes based on need;Accelerated Innovation: open access and collaboration enhance the innovation pipeline.

Open source is not just about accessing code; it is a software engineering paradigm. In tele-exercise, open-source architectures tend to favor microservices-based structures, each with a specific role, such as sensor communication or data processing. Message Queuing Telemetry Transport emerges as an efficient protocol often adopted to ensure communication [18]. The intersection of tele-exercise and open-source technologies crystallizes a unique multidisciplinary synergy. The open-source software architecture is inherently modular, allowing for extensive customization of the user interface (UI). Frameworks such as React or Angular, along with libraries such as Material-UI, can be implemented to provide seamless, responsive user experiences tailored to the user’s needs.

From its origins in the 1970s with the Free Software Foundation and visionaries like Richard Stallman, open source has always promoted the freedom to study, modify and share software. The General Public License is one of the pillars of this philosophy [19]. Another important aspect to consider is the open-source licenses. Licenses define the essence of open source [20]. They guarantee:Use: the absence of restrictions on usage;Access: analysis and auditing through full access to the code;Modifiability: adaptability without boundaries;Distribution: free sharing, regardless of economic model;In the engineering scene, open source is also distinguished by [21];Version Control Systems: Git, as an example, is essential for tracking and managing changes in collaborative development environments;Continuous Integration and Deployment: open source promotes the adoption of Continuous Integration and Continuous Deployment (CI/CD) methodologies, which are essential for constantly updating software;Peer Review: peer review consolidates code quality and adherence to standards.

Open-source libraries and toolkits like A-Frame and AR.js are crucial for creating dynamic virtual training environments. Training sessions can now be adapted in real-time based on the user’s physiological data with the addition of sensors and wearables. In the field of tele-exercise, there is a growing need for real-time data analysis [22]. This has led to the emergence of advanced algorithms that can predict, analyze, and provide performance feedback to users. Platforms such as TensorFlow or PyTorch, being open source, are at the center of this advancement, allowing developers to train artificial intelligence models that can assist users during workouts [23].

The merging of tele-exercise and open source signifies the future of collaboration and innovation, dismantling barriers in technology. This synergy provides fitness enthusiasts with advanced, personalized, and secure training solutions. It also creates new opportunities for collaboration, innovation, and growth in the expanding fitness industry for developers and companies. Overall, it represents a step toward a world where technology and wellness can coexist harmoniously [24].

The digital age has given rise to tele-exercise, with open-source platforms taking a leading role in this field. Open-source architecture not only promotes technological inclusivity but also fosters innovation in tele-exercise. These platforms serve as virtual laboratories for experimentation, offering flexibility through customizable source code. This enables software engineers to tailor the platform to specific needs and target groups. Furthermore, the ability to integrate new algorithms or modules enhances the system’s versatility [25,26]. The modularity of the open-source design also encourages experimentation with advanced sensor systems. The ability to rapidly integrate and calibrate new sensors, ranging from measuring body temperature to sweating and heart rate, expands the range of applicability of tele-exercise, ensuring its relevance in biomechanical and medical research [27,28]. Similarly, the advent of artificial intelligence and Machine Learning has introduced several revolutionary possibilities. AI can analyze in real-time massive datasets derived from tele-exercise sessions. This ability can enable the generation of personalized feedback and suggestions. Moreover, thanks to the power of Machine Learning, platforms can learn and adapt to the specific needs of users, improving the effectiveness of proposed exercises [29]. The collaborative and multidisciplinary environment inherent in open source is another strength: by forming a global network of researchers, developers, and users, an ecosystem of co-creation and interdisciplinarity emerges, where expertise from various branches of engineering and science come together, leading to innovative solutions [30,31].

However, with such opportunities, significant challenges also emerge. Security, for example, is a primary consideration: source code visibility can expose potential vulnerabilities. Although transparency provides system verifiability, protection from external threats requires constant vigilance and the adoption of the latest cybersecurity best practices. In addition, dependence on a development community, the need for adequate technical support, and challenges related to interoperability and project sustainability are other crucial concerns. Microservice-based architectures, common in open-source solutions, provide modularity and scalability. This framework allows for the smooth integration of new or enhanced components without causing systemic disruptions, an essential aspect of maintaining operability and functionality during the research phase. The open-source nature, being often free of onerous licensing costs and having the advantage of an active development community, is a cost-effective option for researchers.

Another important aspect is the speed with which innovation can be implemented. Collaboration among developers globally accelerates innovation, including prototyping, troubleshooting, and integration. Open code allows for the independent verification of functionality, security control, and benchmarking system performance. However, the adoption of open source also presents engineering challenges. While transparency and verifiability are advantages, they can also expose critical details to attackers. Robust security mechanisms, leveraging advanced techniques such as static code analysis and dynamic vulnerability scanning, are critical. Another point of concern is technical support. The lack of guarantees on support and updates can be a barrier for organizations without in-house expertise. Additionally, interoperability between different solutions, while a premise of open source, can present challenges when integrating components from projects with divergent visions or standards. Finally, it is crucial to consider the long-term sustainability of open-source projects. Not everyone maintains an active developer base over time, and depending on solutions that may not be updated in the future can present significant risks [32].

Their efficiency lies in their ability to integrate and process users’ biomechanical and physiological data in real-time, thus offering feedback and interventions based on rigorous scientific evidence. The biomechanics of motion are essential to this integration. Kinematic models represent human motion through equations of motion, allowing for detailed movement analysis. These models, in conjunction with force sensors and accelerometers, provide a comprehensive view of muscle tensions and force distribution during physical activity. Optimization algorithms, such as gradient methods, are implemented to suggest corrections, minimize stress, and improve movement efficiency [33]. In order to evaluate aerobic and anaerobic performance, it is essential to monitor metabolism by measuring heart rate and lactate production and providing accurate feedback. Advanced sensors can ensure individual safety by providing thermoregulation during exercise. By integrating these data with dietary information, an optimal nutritional regimen can be proposed. Neuroscience plays a key role, especially in terms of neuromuscular feedback. Electromyography assesses muscle activation, detecting imbalances or asymmetries, while tele-exercise platforms can incorporate motor learning techniques based on multisensory stimulation and accelerating rehabilitation. Looking ahead, all the technologies discussed above promise a revolution that places the user at the center of their wellness journey, shifting the approach from reactive to proactive.

### Research Purposes

The purpose of this narrative review is to analyze the scientific literature on the application of the aforementioned technologies in the training of healthy individuals. This revolution in tele-exercise, utilizing emerging technologies and open-source platforms, goes beyond fitness and contributes to health promotion and physical awareness education. At present, though the benefits of tele-exercise for patients with pathologies and injuries are already well known by the scientific community, there is a serious lack of scientific studies aimed at defining, regulating and assessing the training parameters and regimes for healthy people. In this analysis, tele-exercise will be presented as a tool no longer for the exclusive use of pathological and rehabilitative situations but as a reality applicable to the general population. During the COVID-19 pandemic, the importance of tele-exercise and alternative healthcare has been widely recognized. Various comprehensive reviews have delved into the role of tele-exercise in healthcare, but their scope has been limited due to the scarcity of randomized controlled trials [34]. For instance, Jassil et al. (2022) [19] conducted a comprehensive review and meta-analysis that encompassed both randomized and non-randomized controlled trials, with a specific focus on single-arm trials. The challenge in conducting a scoping review is the limited number and diverse types of studies being conducted. As a result, our research team aimed to highlight tele-exercise by explaining how it interacts with today’s available technologies.

## 2. Methods

The present narrative review sourced scientific articles for analysis using the PubMed search engine. The search, conducted on 30 May 2023, was based on keywords such as “tele-exercise”, “telerehabilitation”, “remote exercise” and “telehealth exercise” along with “physical activity” or “exercise”. Articles published within the past ten years were carefully scrutinized to identify patterns and trends in the use of tele-exercise. Papers were selected if a study appeared within them that involved any administration of physical activity conducted telematically and not concurrently between instructors and participants. For each clinical trial, the analysis included the type of tele-exercise activity performed (synchronous, asynchronous, or both), the health status of the participants, the platform used for the tele-exercise sessions, the age range of the study participants, and the study duration.

## 3. Results

The analysis conducted on PubMed identified 44 clinical trials [Figure 1], listed in Appendix A. Interestingly, earlier studies predominantly utilized asynchronous modes, whereas more recent ones have shown a preference for synchronous modes. The utilization of a dual mode has been rarely explored in studies. The shift towards synchronous modes can be attributed to technological advancements and increased access to advanced telecommunication technologies. The majority of the clinical trials focused on individuals with various diseases, such as respiratory diseases, cardiovascular diseases, and musculoskeletal diseases, with only a small number (4) involving healthy participants. These findings bear significance in exploring the potential applications of tele-exercise. In light of these considerations, we focused our attention on those few studies that involved only healthy people to highlight their results.

### Analysis of Studies on Healthy People

The first analyzed study aimed to investigate the effects of a synchronous tele-exercise program on physical fitness, quality of life, loneliness, and mood changes in older people experiencing social isolation during the COVID-19 pandemic. The study by Alpozgen and collaborators involved 30 participants, randomly divided into a study group (SG) and a control group (CG). The SG participated in a synchronized online exercise program, while the CG was placed on a waiting list [35]. The results showed significant improvements in the physical fitness level, quality of life, and positive emotional status in the SG. Additionally, there was no significant change in loneliness scores in the SG, while loneliness scores worsened in the CG. The study concluded that synchronous tele-exercise could improve physical fitness, maintain quality of life, alleviate loneliness, and improve positive mood in older people. Despite having had a “particular” target audience compared to the general population, added to an equally particular existential period, the benefits of tele-exercise not for purely rehabilitative purposes or directed at combating a specific disease already appear evident. The second study is also an indirect legacy of the 2019 pandemic. The research conducted by Wilke and collaborators was a randomized controlled trial involving 763 participants from various countries. The mode of interaction between instructors and participants was again synchronous. In addition, this study showed that using a synchronous mode may have benefits compared to an asynchronous mode with exercise recordings. The findings suggest that engaging in tele-exercise resulted in notable improvements in moderate and vigorous physical activity when compared to the control group. However, as the study progressed, the positive effects waned, particularly after the shift to prerecorded workouts. While the WHO-5 scale revealed a slight enhancement of mental well-being during the live-streamed phase, no such benefits were observed during the prerecorded phase. At the onset, the tele-exercise group demonstrated better sleep quality, lower anxiety levels, and higher exercise motivation than the control group. Analyzing the negative aspects: the study faced a high dropout rate, with only 46% of participants completing the first four weeks and 30% completing the entire 8-week study. This attrition rate is considered high compared to other studies and may have been influenced by the frequency of assessments and varying local pandemic restrictions (just imagine the combination of home tele-exercise with the eventual end of the restrictions imposed by the pandemic lockdown). In conclusion, this trial demonstrated synchronous tele-exercise was effective in increasing physical activity and improving some health markers, such as mental well-being, anxiety, and sleep quality, during pandemic-related restrictions. However, the benefits were less consistent after the transition to prerecorded workouts, with maintaining participant engagement over time also being a challenge [36]. The most recent study brings the elderly category back into focus. Although this may appear to be a technologically disadvantaged category in the use of tele-exercise, it appears instead to be the group that can benefit the most from a form of enjoying synchronous remote exercise for individuals who, either for logistical reasons or because of difficulties in leaving the vicinity of their homes, would find it difficult to carry out “traditional” exercise sessions. A study led by Stonsaovapak and collaborators has implemented tele-exercise technology, using wearable sensors, to evaluate and enhance physical performance and walking ability in elderly individuals, as well as decreasing fall risk. This trial was carried out by providing remote exercise programs. The study was conducted in four rural areas of Thailand and involved 123 participants in an 8-week group exercise program. The results showed significant improvements in physical performance and a reduced risk of falls, with high participant satisfaction. The final consideration indicates a potential for remote physical enhancement to increase patient accessibility in rural areas of the Thai healthcare system. This feature could also be extended to other national settings with appropriate adjustments [37]. As the last study, we examined a pre-COVID project that showcased the potential benefits of tele-exercise applications for enhancing the quality of life in healthy individuals. The study, led by Saran and collaborators from various Polish institutions, involved 927 participants of all ages and genders. After using an asynchronous mobile application for remote physical activity monitoring, researchers found that actual physical activity levels were lower despite high self-reported activity. Despite the inaccuracies dictated by the participant’s self-assessment system, this pioneering project has highlighted the potential for readily available devices, such as mobile phones and biomedical devices, to improve habits and promote health. The mobile application provided daily monitoring and motivational feedback to users. This research contributes to our understanding of physical activity patterns and the role of telemedicine in preventing disease, even in healthy individuals [38].

## 4. Discussion

The use of technologies undoubtedly offers many opportunities to maintain people’s wellbeing throughout daily life. Through many platforms, people can track their physical activity (PA) regardless of their location, engaging in PA even when access to gyms or specialized sports facilities is limited. The real-time monitoring of PA and several parameters, such as physical and psychological, offers users instant feedback on their efforts, empowering them to adjust and improve their performance effectively. Furthermore, tele-exercise in healthy people does not lose its ability to be extremely personalized. Despite the fact that more and more people are exercising through tele-exercise and despite its undoubted advantages, it is noteworthy to point out that very little scientific research has focused on tele-exercise in healthy people.

Here, we analyzed four studies on this subject. Alpozgen and collaborators [35] highlight that while loneliness scores did not significantly change in the study group, they worsened in the control group. This finding underscores the potential of synchronous tele-exercise not only in improving physical health but also in addressing emotional well-being and feelings of loneliness among older individuals, so it is possible to conclude that synchronous tele-exercise can lead to improvements in physical fitness, quality of life, loneliness alleviation, and positive mood in older people. These findings suggest that tele-exercise programs can have broader benefits beyond rehabilitative purposes or disease-specific interventions, particularly in times of social isolation and existential challenges, such as those posed by the COVID-19 pandemic. Alongside the pandemic exception, it is possible to think of several other limitations that healthy people may have in terms of doing physical activity, such as limited time to get to and from dedicated exercise facilities, problems with one’s car or transportation and family commitments. The aforementioned study conducted by Stonsaovapak and collaborators [37] highlights the potential of remote physical enhancement programs to improve people’s accessibility to rural healthcare systems, such as those in Thailand. The research provides valuable insights into the potential of tele-exercise technology to support the health and well-being of older individuals, especially in underserved rural areas. Moreover, the research conducted by Wilke and collaborators [36] has shown that engaging in tele-exercise, particularly through synchronous methods, led to significant enhancements in moderate and vigorous PA levels compared to the control group. The study also takes into consideration the psychological sphere, highlighting how in the initial phase of live-streamed workouts there were improvements in mental well-being, sleep quality, anxiety levels, and exercise motivation among participants in the tele-exercise group. Interestingly, when the activity goes from synchronous to asynchronous, positive effects decrease. Indeed, the WHO-5 scale indicated a slight enhancement in mental well-being during the live-streamed phase, which was not sustained during the prerecorded phase.

This result is very important because it stresses the potential differences in outcomes based on the mode of interaction and the delivery of tele-exercise programs. One notable challenge faced in the study was the high dropout rate, with only a portion of participants completing the full duration of the research. Maintaining participant engagement over time proved to be a significant challenge, and it can be inferred that adherence/engagement to the exercise varies depending on whether it is carried out synchronously or asynchronously. The concept of adherence/engagement in relation to PA is quite poor, undefined, and not standardized. Adherence to a program can be interpreted as the proportion of completed sessions compared to the total number of sessions, considering whether the sessions were fully completed or not. For example, a person who exercises three times a week for 30 min per session, following the schedule, would demonstrate 100% adherence. Similarly, a person who trains three times a week but for only 15 min per session would also show 100% adherence if the focus is solely on the number of sessions completed. It is worth noting that the potential benefits of tele-exercise applications for enhancing the quality of life in healthy individuals was also investigated. Saran and collaborators [38] investigated the potential of an asynchronous mobile application for remote PA monitoring, finding that the actual PA levels were lower despite high self-reported activity.

## 5. Conclusions

The realm of tele-exercise is relatively nascent in comparison to the extensive research conducted on conventional physical activity. Consequently, there is anticipation for an increase in research endeavors aimed at optimizing this novel modality. For future research, it is important to investigate several aspects still not fully understood about PA in the healthy population, outlined as follows:The arrival of tele-exercise applications/platforms in large quantities on the open market makes it easy to see how crucial it is to increase the number of studies involving this way of engaging in fitness and exercise. However, many of these applications/platforms used are open-source and not always precisely designed and dedicated for PA. Tele-exercise applications/platforms should be dedicated and specially designed, even on open-source platforms, to ensure maximum security for users’ sensitive data. Moreover, the applications/platforms should not favor a single mode of interaction but rather provide users with a choice. Those who require more interaction, sociability, and real-time guidance can opt for the synchronous mode, while those with less time availability and more commitments can participate through the asynchronous mode [35,36,37,38]. It is also essential to recognize the importance of tele-exercise for healthy individuals as well. While tele-exercise is crucial for people with various ailments, it is noteworthy that trials including people without significant pathologies have also been successful. Studies also indicate that tele-exercise is usable by all age groups when supported by an intuitive platform, offering support for healthier and better conduct in life to both the young and old;It is crucial to further explore the disparities in adherence/engagement levels between PA interventions that are delivered synchronously versus asynchronously. While it is true that asynchronous activity can be carried out at any time of the day, it is also true that synchronous activity ensures many positive aspects, such as the possibility of being corrected in the performance of an exercise, adapting the exercise for contingent needs, and being able to carry out the exercise in a group, thus ensuring one of the basic needs of human beings, namely that of relationships;The researchers are quite divided with regard to the healthy status of the participants involved. Indeed, healthy participants can be sedentary, or moderately or highly active, despite their age, making the collected data difficult to interpret. Particularly active people, for example, might be more likely to follow workout programs or applications/platforms than sedentary people;The quality of both the smart devices and internet connection used is not always described [32]. Although this last point may seem minor, just think of how the meetings or video calls that we all did at least once during these last few years, particularly because of COVID-19, were frustrating when the connection was poor. Poor connection, indeed, can lead to abandonment of the exercise, as the activity becomes blocked or poorly seen, and this point should be always investigated in the future research in this field;It is necessary to keep in mind that, for some people, using technology to do PA can even lead to the opposite effect, which is to develop an addiction.

We therefore encourage collaborators to consider all the as-yet unresolved issues we have highlighted, as the field of research on tele-exercise in the healthy population is almost completely unexplored and in need of further study, as well as suggestions to practitioners and guidelines. The increasing prevalence of online trials and exercise protocols carried out on healthy subjects could provide a solid foundation for a comprehensive scoping review (one lacking as of now). As technology continues to advance, tele-exercise is expected to gain traction and attract a massive number of practitioners. Understanding its various dimensions and its integration into the daily routines of the practitioners of physical activity is essential.

### Limitations of the Study

The current review has brought to light a crucial issue in the existing literature: the lack of scientific studies that focus on defining, assessing, and thoroughly describing training parameters and regimens for individuals without health issues. This research gap inhibits our ability to understand the best training practices for healthy individuals and underscores the necessity for further investigation in this area. While the existing literature recognizes the advantages of tele-exercise for specific populations, such as those affected by pathologies, there is a noticeable shortage of research aimed at establishing concrete guidelines and frameworks tailored for the general, healthy population. The scientific articles were sourced from PubMed, which we consider to be a highly regarded database. As we move forward, it would be advantageous to broaden our research to include other scientific databases to gain a more comprehensive understanding of the subject.

## Figures and Tables

**Figure 1 healthcare-12-01466-f001:**
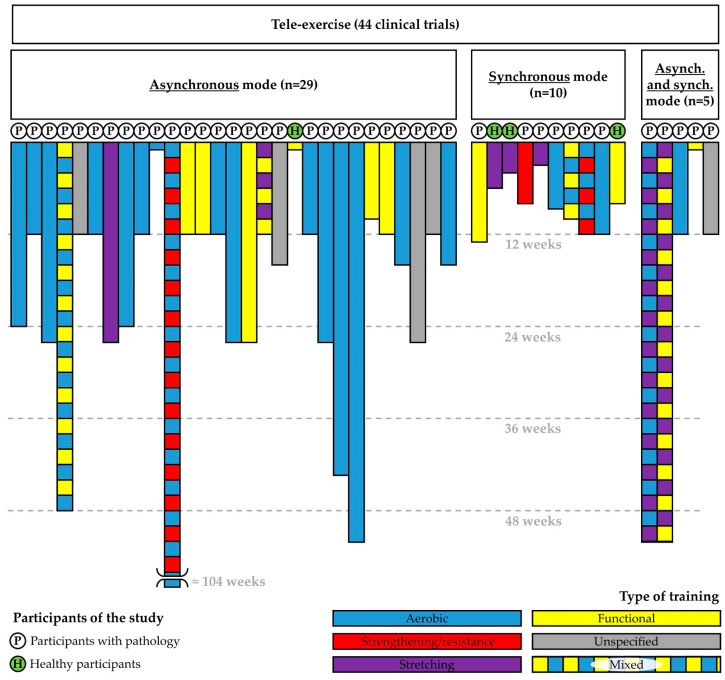
Graphical representation of the clinical trial using tele-exercise found on PubMed. The colors used to represent the mixed trainings indicate the types of training: for example, a mixed training represented using blue and yellow colors indicates that this type of mixed training was composed of aerobic and functional training.

## Data Availability

Data sharing is not applicable, no new data were created or analyzed in this study.

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
