# Peer review of "Emerging Technologies and Open-Source Platforms for Remote Physical Exercise: Innovations and Opportunities for Healthy Population—A Narrative Review"

_healthcare, 2024, doi:10.3390/healthcare12151466_

Round 1
Reviewer 1 Report
Comments and Suggestions for Authors
I believe it's important to address the following points:
1. In the abstract and later in the methodology section, it is crucial to detail the specific criteria used to select the 44 research studies.
2. It's essential to ensure precision in the data presented in scientific research; therefore, the use of vague terms like "10%" should be avoided in favor of accurate data.
3. The authors should have provided an explanation for why they opted not to utilize other databases when they only had access to 4 studies, especially since they acknowledged this limitation in the study. This leads to concerns about the robustness of the study's conclusions.
4. The authors effectively introduced the fundamental concepts related to the problem in the introduction, using relevant references for support.
5. The recent references used in the paper, both in the introduction and the discussion, are praiseworthy.
6. The results section should provide further elaboration on the different modes identified in the analysis of 44 clinical trials on PubMed, for the benefit of the readers.
Author Response
Thanks to suggestions received from the reviewer, we have included the keywords used to search for the articles within the abstract to facilitate understanding of the methodology applied at that stage of the work. Regarding the selection made explicit within the methodology, we have added an additional criterion that we imagined was underneath but actually it was better to make it clear to the reader. We correctly entered the number of studies involving healthy subjects versus the total number of papers and entered the final percentage value. Regarding the exclusive repurposing of results related to PUBMED as a database: we had made a preliminary search of several databases containing research papers, but in addition to an overlap of papers, the others appeared more numerically deficient without any additional studies or trials that could be included. Therefore, to facilitate the reading of the article, in addition to the analytical work, we preferred to refer to a solid and valid database containing the largest number of papers.
We greatly appreciate the interest in our article and the constructive criticism that has been given, and we believe it has been instrumental in improving our work.
Reviewer 2 Report
Comments and Suggestions for Authors
First of all, I would like to thank the authors for their overall efforts during the study. It is a good and clearly described study. The topic is interesting, yet, in its current form, this paper cannot be considered for publication. However, I see value in the research approach and strongly encourage the authors to address the following points.
Abstract
· Please provide more detailed information about the methods in the Abstract.
· Line 18: Please add "database" after PubMed.
Introduction
·The introduction effectively introduces the topic, but consider consolidating several paragraphs that address similar contexts to improve readability.
·Line 41: Add a reference after the first sentence to strengthen the argument.
· Line 47: Transition more smoothly when discussing the shift to wearable devices.
·Line 55: Include additional citations to support the statement that "VR can assist rehabilitation."
· Lines 104-107: Provide strong citations to support this section.
·Lines 125-128: Ensure these statements are well-supported with citations.
· Lines 167-168: Avoid repetition of similar statements found earlier in the text.
·Line 198: Remove "RCTs" as it is only used here; review and remove other unnecessary abbreviations throughout the text.
Methods
· Provide more detailed information in the Abstract section, as part of the narrative review.
Results
·Present the main results obtained from the study rather than overly detailed findings.
·Remove personal thoughts and information that could be more appropriately placed in the Introduction.
· Identify parts that could be integrated into the Discussion section.
Discussion and Conclusions
· The discussion and conclusions are well-structured and well-written.
·Consider removing or reducing the first paragraph, as it does not introduce new information beyond the Introduction.
· Briefly explain the study aims and main findings instead.
· Review and organize the reference list. It seems messy.
Comments on the Quality of English LanguageMinor editing of English language required.
Author Response
Regarding the constructive suggestions provided to us by the reviewer:
We have added more details about the research methodology already in the abstract as well as in the methodology section; as pointed out to us by multiple reviewers we decided to merge and consolidate the different paragraphs that make up the introductory part in order to facilitate reading. We have included the required citations that actually help in reinforcing the concepts as well as supplemented with more recent work. In details we added reference, as requested, in lines: 41, 55, 104-107, 125-128
Any repeated concepts were removed in order to streamline the work and increase reading fluency, as well as any unrepeated abbreviations.
We have rewritten and shortened the introductory part of the results so that the key studies are presented immediately. Discussion and conclusions were separated (as requested by other reviewer) but the concepts expressed were not altered.
Sincere thanks for highly supportive criticism the suggestions that have enabled us to improve our work considerably.
Reviewer 3 Report
Comments and Suggestions for Authors
- There are too many short paragraphs in the introduction (Lines 77-81, 98-102, 112-117, 118-123, 125-128, 129-134, 135-139, 185-186, and 204-205). A paragraph should consist of several sentences dealing with one topic. However, short paragraphs reduce readability. Add content to each paragraph or combine them with others.
- Is Figure 1 a result? Please add a citation (Figure 1) at the end of the sentence that refers to this content in the results section.
- Do not just divide the results into paragraphs, but use subheadings to separate the content.
- Separate "4. Discussion and Conclusions" into two sections. Write "5. Conclusions" clearly.
Author Response
Regarding the constructive suggestions provided to us by the reviewer:
We reshaped the text, moving, editing and adding in order to file down the number of short paragraphs present. Effectively now the work appears smoother and reading is made easier.
The citation to Figure 1 has been included within the text.
Added subsection in results regarding analysis of studies conducted on healthy subjects.
Clearly and distinctly separate discussions from conclusions.
Reviewer 4 Report
Comments and Suggestions for Authors
The authors present interesting findings of virtual exercise programs (live, recorded, and mixed) in healthy populations. They do not conduct a systematic review with detailed methodology, a priori hypotheses, or PICOS questions, which this reviewer feels would have enhanced this manuscript proposal. There is also a lot of "information" about the relevant technologies, which this reviewer feels would be better presented as a separate "background" manuscript proposal--a kind of special article, technology update, or letter. The new findings summarized from the studies the authors found in their literature review also would be best presented in a short report or letter. In the manuscript proposal's current form, it is too long and rambling, which is counterproductive in getting across to the reader the seemingly very valid and helpful aspects of the work. This reviewer would encourage the authors to rewrite the manuscript proposal as two shorter pieces, as described above, and resubmit these two pieces.
Comments on the Quality of English LanguageGood proofreading should also be carried out, though this reviewer does not recall specific concerns.
Author Response
Thanks for the valuable suggestions:
Having only later received this revision, we can say that the work, also thanks to the sharp and relevant comments of the other reviewers, has been reshaped. We have trimmed down some of the more difficult parts, tried to make some sections more reader-friendly, removed sub-sections, and moved topics around. At the end of this massive editing operation, we can say that the work has indeed been improved. In earnest, we opine that the segmentation of work into a distinctly 'technological' domain and a dedicated telematics-based exercise section would diminish cohesion. It is our assertion that these two segments are inherently interconnected, and it is essential to underscore that one serves as the foundation for the other. Therefore (as has also been pointed out to us), the combination of the two sections appears to provide a plus to the work, especially now that many critical 'technological' terminology barriers have been removed. We therefore ask that the suggestions received be revised in the light of the new version of the submitted work,
Cordially,
Round 2
Reviewer 1 Report
Comments and Suggestions for Authors
Dear authors, I have reviewed your corrections resulting from my suggestions and consider that the paper can now be approved for publication in this journal.
Author Response
Comment: Dear authors, I have reviewed your corrections resulting from my suggestions and consider that the paper can now be approved for publication in this journal.
Thank you for your valuable advice, we are happy that the revision work done was appreciated.
Reviewer 3 Report
Comments and Suggestions for Authors
All previously provided comments have been adequately addressed.
Thank you for your hard work on the research and writing of this paper.
Author Response
Comment:
All previously provided comments have been adequately addressed.
Thank you for your hard work on the research and writing of this paper.
Response: Thank you for your valuable advice, we are happy that the revision work done was appreciated.
Reviewer 4 Report
Comments and Suggestions for Authors
While the authors heavily revised their existing manuscript, this review does not feel the authors present enough new information to justify this type of manuscript proposal. The topic is timely and relevant, though. A letter or other short report on emerging technology seems more appropriate... a very short essay of 300-500 words communicating only essential high points. This reviewer understands this may be difficult but could result in a much stronger manuscript proposal with higher impact. This reviewer was in this exact situation, and the outcome was indeed very good.
Comments on the Quality of English LanguageGood proofreading should also be carried out, though this reviewer does not recall specific concerns.
Author Response
Comment: While the authors heavily revised their existing manuscript, this review does not feel the authors present enough new information to justify this type of manuscript proposal. The topic is timely and relevant, though. A letter or other short report on emerging technology seems more appropriate... a very short essay of 300-500 words communicating only essential high points. This reviewer understands this may be difficult but could result in a much stronger manuscript proposal with higher impact. This reviewer was in this exact situation, and the outcome was indeed very good.
Response: The idea of a short communication had also come to our mind, but in the end we thought that that solution is intended for a different type of reader and that we would prefer to deepen the discourse by integrating the overall vision of two realities that may appear distant from each other but which instead support each other. We promise ourselves, in the future, at the end of the project we are working on, to also create a short communication in order to more 'easily' introduce a larger number of readers to the world of programming, apps and tele-exercise. In this regard, we ask the reviewer to contact us if he would like to participate, we could create something together.